# ADVANCING VISION TRANSFORMERS WITH GROUP-MIX ATTENTION

## ABSTRACT

Vision Transformers (ViTs) have shown to enhance visual recognition through modeling long-range dependencies with multi-head self-attentions (MHSA), which is typically formulated as Query-Key-Value computation. However, the attention map generated from the Query and Key only captures token-to-token correlations at one single granularity. In this paper, we argue that self-attention should have a more comprehensive mechanism to capture correlations among tokens and groups (i.e., multiple adjacent tokens) for higher representational capacity. Thereby, we propose Group-Mix Attention (GMA) as an advanced replacement for traditional self-attention, which can simultaneously capture token-to-token, token-to-group, and group-to-group correlations with various group sizes. To this end, GMA splits the Query, Key, and Value into segments uniformly and performs different group aggregations to generate group proxies. The attention map is computed based on the mixtures of tokens and group proxies and used to re-combine the tokens and groups in Value. Based on GMA, we introduce a powerful backbone, namely GroupMixFormer, which achieves state-of-the-art performance in image classification, object detection, and semantic segmentation with fewer parameters than existing models. For instance, GroupMixFormer-L (with 70.3M parameters and $384^2$ input) attains 86.2% Top-1 accuracy on ImageNet-1K without external data, while GroupMixFormer-B (with 45.8M parameters) attains 51.2% mIoU on ADE20K. Codes and trained models will be released.

## 1 INTRODUCTION

Vision Transformers (ViTs) significantly improve visual recognition tasks, including image classification (Dosovitskiy et al., 2021; Yuan et al., 2021), self-supervised learning (Chen et al., 2021d; Caron et al., 2021; Xie et al., 2021b; Bao et al., 2021), object detection (Liu et al., 2021b; Dai et al., 2021), and semantic segmentation (Wang et al., 2021; 2022; Xie et al., 2021a). One crucial module that contributes significantly to the performance improvement is the multi-head self-attention (MHSA), which enables network designing with the long-range dependency modeling (Vaswani et al., 2017; Raghu et al., 2021), global receptive field, higher flexibility (Jia et al., 2021; Cordonnier et al., 2019) and stronger robustness (Paul & Chen, 2021; Xie et al., 2021a). Typically, the term "attention" (i.e., the Q-K-V attention) means linearly re-combining *Value* with the correlations between the *Query* and *Key*, which are usually computed between pairs of individual tokens.

However, it's empirically found that there is a major limitation in Q-K-V self-attention, which is shown in Figure 1: the attention map only describes the correlations between each individual token pairs at one single granularity (Figure 1(a)), and multiplying the attention map with the Value only linearly re-combines the individual tokens. This framework obviously does not consider the correlations among different token groups (i.e., neighborhoods) at various granularities. For one specific example, self-attention does not correlate the nine tokens at the top-left corner as a whole to those groups at the bottom-right. This limitation, though obvious, has been unintentionally neglected because the Q-K-V computation seems to be capable enough of modeling the mappings from input to output, as any entry in the output attends to each individual entry in the input.

In this study, we propose a more comprehensive modeling approach, referred to as the *Group-Mix Attention* (**GMA**), to alleviate the aforementioned limitations of the widely used Q-K-V self-attention mechanism. GMA splits the tokens into uniform and distinct segments and substitutes some individual

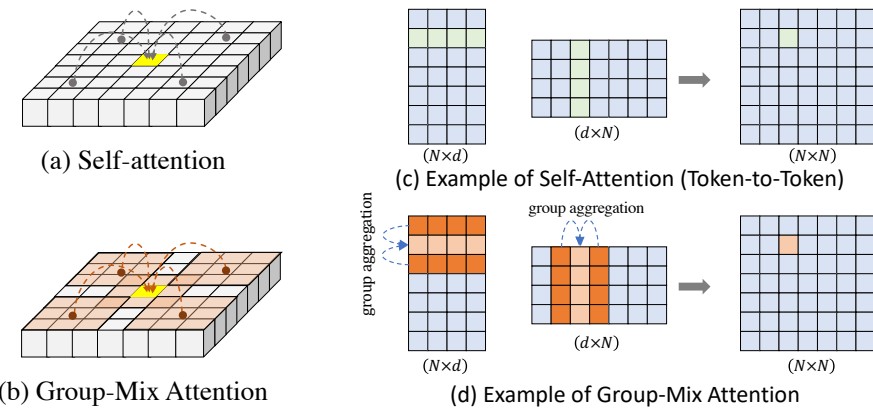

(a) Self-attention

(c) Example of Self-Attention (Token-to-Token)

(b) Group-Mix Attention

(d) Example of Group-Mix Attention

Figure 1: **Conceptual comparisons between the self-attention and Group-Mix Attention (GMA).** In (a) and (b), we showcase with 7×7 single-dimensional tokens. Unlike the self-attention that computes correlations between pairs of individual tokens, GMA creates proxies of token groups (e.g., nine adjacent tokens) via group aggregators, and then computes the group-to-group correlations via proxies. In (c) and (d), we show the concrete computation of GMA with seven four-dimensional tokens, so that $N$=7 and $d$=4. To compute the correlations between two highlighted groups that each consist of three tokens, we aggregate them into two proxies for further multiplication. The group aggregation can be effectively implemented via sliding-window-based operators.

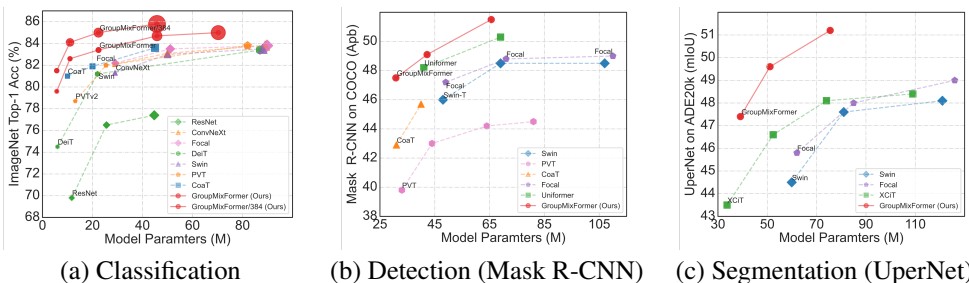

(a) Classification    (b) Detection (Mask R-CNN)    (c) Segmentation (UperNet)

Figure 2: **Performance of GroupMixFormer compared to the state-of-the-art models.** We evaluate GroupMixFormer on standard benchmarks, including classification on ImageNet-1K (Russakovsky et al., 2015) without extra data in (a), object detection on COCO (Lin et al., 2014) in (b), and semantic segmentation on ADE20K (Zhou et al., 2019) in (c). The computational complexity is denoted as the geometry area. GroupMixFormer performs favorably against ViT and CNN models including DeiT (Touvron et al., 2021), Swin (Liu et al., 2021b), PVT (Wang et al., 2021), CoaT (Xu et al., 2021), Focal (Yang et al., 2021), ConvNeXt (Liu et al., 2022), etc.

tokens with group proxies generated via group aggregators, as shown in Figure 1 (b). Afterward, we compute the attention map with the Query and Key (where some tokens have been replaced by group proxies) and use it to re-combine both the group proxies together with individual tokens in Value. The proposed GMA has some appealing advantages: **(1)** GMA is capable of modeling correlations among not only individual tokens but also groups of tokens. Different kinds of attentions are *mixed* to obtain a better understanding of the tokens from a comprehensive aspect. The token-to-token, token-to-group, and group-to-group correlations are simultaneously modeled for higher representational capabilities. **(2)** GMA is efficient and easy to implement. The group-to-group correlation is computed via aggregating the groups into proxy tokens and then computing the correlation between proxies (as shown in Figure 3). Such a process can be efficiently implemented with sliding-window-based operations, e.g., pooling and convolution.

Building on GMA, we develop a hierarchical vision transformer, GroupMixFormer, which can serve as visual backbones for various tasks. We evaluate GroupMixFormers on standard visual recognition tasks, including image classification, object detection, and semantic segmentation, and conduct comparisons with advanced models as shown in Figure 2. Our results demonstrate the effectiveness of our designs. For example, a small GroupMixFormer instance (with 22.4M parameters) achieves 83.4% Top-1 accuracy on ImageNet-1K, comparable to the much larger Swin-B(Liu et al., 2021b) (88M

parameters). Additionally, GroupMixFormer also performs favorably against state-of-the-art ViTs and CNNs on object detection and semantic segmentation. On the ADE20K dataset, GroupMixFormer-B achieves 51.2% mIoU with a backbone size of 46M. Extensive experiments also demonstrate that effectively modeling the correlations among tokens and diverse groups is crucial for the success of GMA. Such a design paradigm can also be readily adopted into other ViT architectures as an advanced replacement for traditional self-attention.

## 2 RELATED WORKS

### 2.1 VISION TRANSFORMER

Vision Transformer (ViT) (Dosovitskiy et al., 2021) first introduces the Transformers into computer vision. Unlike CNN-based architectures, ViT utilizes sequentially-connected Transformer encoders (Vaswani et al., 2017) on the visual token sequence. The multi-head self-attention (MHSA) mechanism employed in ViTs captures global dependencies effectively, giving them an edge over CNN neural networks (He et al., 2016; Huang et al., 2017) in both supervised (Graham et al., 2021; Touvron et al., 2021) and self-supervised scenarios (Chen et al., 2021d; Caron et al., 2021). To advance the general performance of ViTs, a series of researches have been conducted, including data-efficient training (Touvron et al., 2021), token re-designing and selection (Rao et al., 2021; Liang et al., 2022), pyramid structures (Liu et al., 2021b; Wang et al., 2021), modulation on self-attention mechanism (Zhang et al., 2021; Chen et al., 2021a;c), etc. Most of these works adopt the original Q-K-V computation, which is found to be effective in processing visual information. In this work, we aim to further advance the general performance of ViTs by introducing Group-Mix Attention (GMA). Unlike prior arts, GMA is capable of modeling the correlations among not only individual tokens but also groups of tokens, thus leading to comprehensive representational capabilities.

### 2.2 COMPREHENSIVE MODELING OF SELF-ATTENTION

To enhance representational abilities of self-attention, several approaches have been explored from different perspectives. 1) Introducing locality has proved effective, as evidenced by the Swin Transformers (Liu et al., 2021b;a) and Focal Transformer (Yang et al., 2021), which conduct attention computation within local windows. 2) Computing correlations with pre-defined patterns can enhance the capability of self-attention, as demonstrated by the CSWin Transformer (Dong et al., 2021) and Pixelfly-Mixer (Chen et al., 2021b), both of which attempt to compute attention with pre-defined and carefully-designed patterns to realize more comprehensive modeling. 3) Other network architectures (Xu et al., 2021; Lee et al., 2021; Wang et al., 2022; Li et al., 2022) have also been combined to create more comprehensive models. In this work, we focus on the limitations caused by token-to-token correlations at one single granularity and propose an advanced attention mechanism (i.e., **GMA**) that constructs a more comprehensive prototype of self-attention, which clearly distinguishes our method from previous approaches.

## 3 GROUPMIX ATTENTION AND GROUPMIXFORMER

We introduce the motivation behind the high-level idea in Section 3.1, elaborate on the structural designs in Section 3.2, and describe the architectural configurations in Section 3.3.

### 3.1 MOTIVATION: FROM INDIVIDUAL TO GROUPS

We discuss the limitations of self-attention starting from its vanilla form. Let $X \in \mathbb{R}^{N \times d}$ be the input tokens, where $N$ is the token number and $d$ is the dimension. The output of vanilla self-attention is,

$$Y = \text{Softmax}(XX^T)X. \tag{1}$$

Note that we ignore the normalization factor $\frac{1}{\sqrt{d}}$ for brevity. Intuitively, by the definition of matrix multiplication, $XX^T$ calculates the similarity/correlation between each two of the tokens. The output of the softmax function $A \in \mathbb{R}^{N \times N}$ is called an attention map. The multiplication $AX$ means linearly re-combining the tokens according to the attention map at each location.

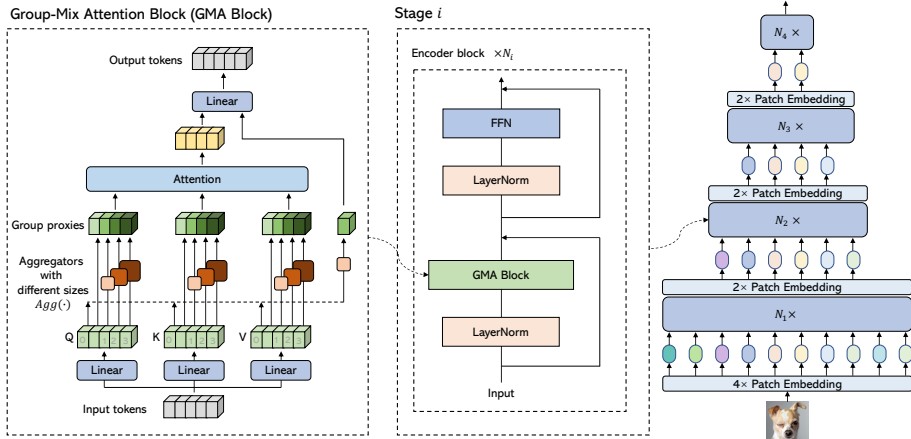

Figure 3: **Structural designs of Group-Mix Attention Block and architecture of GroupMix-Former.** In each GMA block, we split $Q$, $K$, and $V$ into five segments and use aggregators with different kernel sizes to generate group proxies on four of them, so that we can conduct attention computation on mixtures of individual tokens and group proxies of different sizes. The branches whose outputs are fed into the attention computation are referred to as the pre-attention branches. To construct diverse connections, the rightmost branch utilizes aggregation but without attention, which is termed the non-attention branch. A linear mapping layer is adopted to fuse the outputs from the attention and non-attention branch. For clear illustration, we use $\text{Agg}^1$, $\text{Agg}^2$, and $\text{Agg}^3$ in the pre-attention branch to denote the aggregators with kernel sizes of 3, 5, and 7, respectively, and use $\text{Agg}^0$ for the aggregator in the non-attention branch.

We note a limitation of this form. There may exist certain patterns (i.e., group patterns) that require treating some specific tokens as a group with diverse granularities. However, self-attention lacks an explicit mechanism for modeling such patterns, as it only considers correlations between pairs of individual tokens at a single granularity (i.e., individual patterns). In this paper, we seek to utilize both individual patterns and group patterns for comprehensive modeling. For group patterns, we seek to correlate some neighborhoods of tokens to the other neighborhoods. This paper proposes to realize this by generating *group proxies* in Query, Key, and Value, and performing the Q-K-V computation with proxies, which is described in Section 3.2. We experimentally found that explicitly modeling the correlations among groups with diverse sizes and individual tokens significantly improves the performance of not only the proposed GroupMixFormer but also other ViTs with different attention modules (e.g., Swin Transformer (Liu et al., 2021b) and PVT (Wang et al., 2021), as shown in Table 8), demonstrating that upgrading the fundamental component can benefit multiple ViTs.

## 3.2 GMA: MIXING GROUPS FOR BETTER ATTENTION

We introduce GMA to model the group patterns as aforementioned. In GMA, we generate the group proxies by simply replacing some entries in the Query, Key, and Value with aggregations of some whole groups, which can be efficiently implemented with sliding-window-based operations $\text{Agg}(\cdot)$, such as max-pooling and convolution. Typically, the Q/K/V entries are uniformly divided into $n$ segments and we perform aggregation on some segments. Without loss of generality, we use $\text{X}_i$ ($i \in [1, \cdots, n]$) to denote one segment (X may represents Q, K, or V) and the aggregations as $\text{Agg}^i(\text{X}_i)$. Note that the aggregator may be different for each segment. To perform attention computation, we concatenate the aggregations $\text{Agg}^i(\text{X}_i), i \in [1, \cdots, n]$ to produce $X'$. In this way, we obtain group proxies $Q'$, $K'$, and $V'$. Afterward, we perform attention computation as introduced in (Xu et al., 2021; Ali et al., 2021; Shen et al., 2021) on the group proxies to generate the output.

Note that we maintain the feature resolution after aggregation. Without reducing the spatial resolution, GMA brings fine-grained features for attention computation, which outperforms those with decreased feature sizes (Fan et al., 2021; Wu et al., 2021). We use depth-wise convolutions with various kernel sizes to implement aggregators $\text{Agg}(\cdot)$. Note that as the inputs of attention are now group proxies, we achieve correlating K×K tokens simultaneously (K denotes the kernel size of $\text{Agg}(\cdot)$, which may be different for each segment) instead of individual tokens, which is more sufficient and comprehensive for modeling correlations.

Table 1: **Architectural configurations of GroupMixFormer models.** We use D, R, and L to denote the dimension of tokens, the expansion ratio of FFN, and the number of encoder blocks. We use M/T/S/B/L (mobile/tiny/small/base/large) to label models with different scales.

| | Output size | GroupMixFormer -M(5.7M) | GroupMixFormer -T(10.9M) | GroupMixFormer -S(22.4M) | GroupMixFormer -B(45.8M) | GroupMixFormer -L(70.3M) |
|---|---|---|---|---|---|---|
| stage 1 | $\frac{H}{4} \times \frac{W}{4} \times D_1$ | $D_1 = 40$ $R_1 = 4, L_1 = 3$ | $D_1 = 80$ $R_1 = 4, L_1 = 4$ | $D_1 = 80$ $R_1 = 4, L_1 = 2$ | $D_1 = 200$ $R_1 = 2, L_1 = 8$ | $D_1 = 240$ $R_1 = 4, L_1 = 8$ |
| stage 2 | $\frac{H}{8} \times \frac{W}{8} \times D_2$ | $D_2 = 80$ $R_2 = 4, L_2 = 3$ | $D_2 = 160$ $R_2 = 4, L_2 = 4$ | $D_2 = 160$ $R_2 = 4, L_2 = 4$ | $D_2 = 240$ $R_2 = 2, L_2 = 8$ | $D_2 = 320$ $R_2 = 4, L_2 = 10$ |
| stage 3 | $\frac{H}{16} \times \frac{W}{16} \times D_3$ | $D_3 = 160$ $R_3 = 4, L_3 = 12$ | $D_3 = 200$ $R_3 = 4, L_3 = 12$ | $D_3 = 320$ $R_3 = 4, L_3 = 12$ | $D_3 = 320$ $R_3 = 2, L_3 = 16$ | $D_3 = 360$ $R_3 = 2, L_3 = 18$ |
| stage 4 | $\frac{H}{32} \times \frac{W}{32} \times D_4$ | $D_4 = 160$ $R_4 = 4, L_4 = 4$ | $D_4 = 240$ $R_4 = 4, L_4 = 4$ | $D_4 = 320$ $R_4 = 4, L_4 = 4$ | $D_4 = 480$ $R_4 = 2, L_4 = 16$ | $D_4 = 480$ $R_4 = 2, L_4 = 16$ |

The idea of using sliding-window-based operations to aggregate groups into proxies, though simple, is the key to the mechanism of mixing groups of *different sizes* and individual tokens at *various granularities*, as we use a different kernel size of aggregator for each segment. Such a process can be efficiently implemented via splitting segments, feeding them through aggregators implemented with different kernel sizes, and concatenating the outputs. Moreover, we employ an identity mapping on one segment instead of an aggregator to maintain the network's abilities in modeling individual token correlations. Therefore, we can model correlations among both groups and tokens while computing the attention map. Multiplying the attention map with the Value can be viewed as re-combining the corresponding groups together with individual tokens accordingly.

Specifically, following the implementation of self-attention (Shen et al., 2021; Ali et al., 2021; Xu et al., 2021), we also use three learnable linear projections to generate Q, K, and V. Afterward, we split Q/K/V uniformly into five segments, which each participates in different computations. As shown in Figure 3 (the left part), a branch corresponds to an aforementioned segment, and the four branches whose outputs are fed into the attention computation are referred to as the pre-attention branches. In three of the pre-attention branches, we use various implementations (e.g., min-pooling, avg-pooling, max-pooling, depth-wise convolution) as the aggregator Agg(·) with different kernel sizes, which are set as 3,5,7, respectively. The results in Table 6 indicate that each of these implementations achieves favorable performance, which shows that aggregation is a crucial step for attention advancement while its implementation can be flexible. We adopt the depth-wise convolutions, whose results are slightly better, in our paper. We further diversify the structures by using no aggregator in the last pre-attention branch, making it an identity mapping. Apart from such a branch with attention but no aggregator, we construct another branch with an aggregator but no attention, which is referred to as the non-attention branch. Finally, the outputs are mixed by a token ensemble layer, which is simply implemented by a linear projection with normalization (Ba et al., 2016) and activation.

## 3.3 ARCHITECTURAL CONFIGURATIONS

Building on the proposed Group-Mix Attention, we introduce a series of vision Transformers named GroupMixFormer, as shown in Figure 3. We adopt a hierarchical (Liu et al., 2021b; Wang et al., 2021) topology with four stages. The first 4× patch embedding layer embeds images into tokens, which is implemented with two sequential 3×3 convolutional layers, each with a stride of 2 and another two 3×3 layers with a stride of 1. At the beginning of each last three stages, we use a 2× patch embedding, which is also implemented with a 3×3 convolution. Within each stage, we construct several encoder blocks. Apart from a GMA block introduced in the last subsection, an encoder block also contains a Feed-Forward Network (FFN), Layer Normalization (Ba et al., 2016) and identity shortcuts, following the common practice in (Dosovitskiy et al., 2021; Touvron et al., 2021; Liu et al., 2021b; Wang et al., 2021; Yang et al., 2021). For image classification, the final output tokens are fed into the classifier after global average pooling (GAP); for dense prediction tasks (e.g., object detection and semantic segmentation), the task-specific heads can utilize the pyramid features output by the four stages. We do not adopt positional encoding in our model since we have naturally broken the permutation invariance with the GMA aggregators.

We instantiate four models with different architectural configurations. The architectural hyper-parameters include the number of encoder blocks in each stage $L$, the embedded dimension $D$, and the MLP ratio $R$, as shown in Table 1. Following the prior works (Wang et al., 2021; Liu et al., 2021b; Touvron et al., 2021), our models scale up from the mobile-scale GroupMixFormer-M (5.7 M) to the large-scale GroupMixFormer-L (70.3 M).

## 4 EXPERIMENTS

In this section, we evaluate our GroupMixFormer on standard visual recognition benchmarks including ImageNet-1K (Russakovsky et al., 2015), MS-COCO (Lin et al., 2014), and ADE20k (Zhou et al., 2019). We present the implementation details for each scenario, quantitative comparisons to state-of-the-art vision backbones, and ablation studies in the following.

### 4.1 IMPLEMENTATION DETAILS

We evaluate the image classification performance of GroupMixFormer on the ImageNet-1K dataset. We follow (Zhang et al., 2017; Yun et al., 2019; Touvron et al., 2021) to augment data and use the training recipe in (Liu et al., 2021b). We train GroupMixFormer for 300 epochs using an initial learning rate of $10^{-3}$ with a 20-epoch linear warm-up. AdamW optimizer (Loshchilov & Hutter, 2017) is utilized with a weight decay of 0.05 and a cosine learning rate schedule. The stochastic depth drop rates (Huang et al., 2016) are set to 0.0, 0.1, 0.2, 0.4, and 0.5 for GroupMixFormer-M/T/S/B/L, respectively. For higher resolutions (e.g., $384^2$ or $448^2$), we finetune the models in another 30 epochs with the learning rate initialized as $2 \times 10^{-6}$ and a linear warm-up for 5 epochs. The finetuning process uses AdamW (Loshchilov & Hutter, 2017) with a weight decay of $10^{-8}$ for optimization.

For object detection and instance segmentation, COCO 2017 dataset is utilized. Specifically, we employ GroupMixFormer as the backbones of Mask R-CNN (He et al., 2017) for object detection and segmentation, and RetinaNet (Lin et al., 2017) for detection only. All the backbones are initialized via the corresponding ImageNet pretrained models. We follow the training schedules in (Chen et al., 2019): the initial learning rate is set to $10^{-4}$ with a linear warm-up for 500 iterations and gradually Wdecreases to $10^{-5}$ and $10^{-6}$ at the 24-*th* and 33-*th* epochs, respectively. We use AdamW (Loshchilov & Hutter, 2017) for both Mask R-CNN and RetinaNet, but the weight decay is 0.05 for the former and $10^{-4}$ for the latter. Except for COCO, we also evaluate the semantic segmentation performance on ADE20k with UperNet (Xiao et al., 2018) and Semantic FPN (Kirillov et al., 2019). We follow (Wang et al., 2021; Liu et al., 2021b) to use the public toolkit (Contributors, 2020) for training and evaluations. The Semantic FPN is trained for 80k iterations, while the UperNet is trained for 160k iterations, both with an AdamW optimizer.

### 4.2 COMPARISONS WITH STATE-OF-THE-ART MODELS

**Image Classification.** We compare the proposed GroupMixFormer with the state-of-the-art models from the literature in Table 2, where all the reported results use only ImageNet-1k for pre-training. Note that we do not use any extra augmentations, like token-labeling (Jiang et al., 2021), knowledge distillation, SAM (Foret et al., 2020), etc. We observe that GroupMixFormer consistently achieves higher Top-1 accuracies than the ViT and CNN models under similar model sizes and computational complexity constraints. Specifically, tested with a resolution of $224^2$, GroupMixFormer-S yields an accuracy of 83.4% with only 22.4M parameters, significantly outperforming the second best ViT (Focal-Tiny (Yang et al., 2021)) by 1.2% and the best CNN (ConvNext-T (Liu et al., 2022)) by 1.3%. Meanwhile, GroupMixFormer-B trained with $224 \times 244$ images even achieves a similar accuracy with Swin-B (Liu et al., 2021b), even though the size of GroupMixFormer-B is only half as that of Swin-B. Moreover, GroupMixFormer shows satisfying scalability towards higher resolution. For example, finetuning with a resolution of $384^2$ further improves the performance of GroupMixFormer-S to 85.0%; with around 70M parameters, our GroupMixFormer-L achieves 85.0% with a resolution of $224^2$ and 86.2% with $384^2$.

**Object Detection.** Table 3 shows the object detection results on COCO with Mask R-CNN and RetinaNet detectors. With Mask R-CNN, GroupMixFormer achieves higher average precision under similar model parameters. Specifically, GroupMixFormer-T performs 1.0% higher (i.e., 47.5% v.s. 46.5%) than the second-best model, which is CoaT Mini, while maintaining a smaller model size of 30.8 M. Besides, our GroupMixFormer-B achieves an $AP^b$ of 51.5%, surpassing all the comparable models. With RetinaNet, GroupMixFormer also shows superiority: GroupMixFormer-T performs 0.5% better than Swin-B (i.e., 46.3% v.s. 45.8%) though ours is much smaller (i.e., 20.2 M v.s. 98.0 M); GroupMixFormer-B performs 2.9% better (i.e., 50.2% v.s. 47.3%) than the Focal-small model. These results show that GroupMixFormer achieves favorable performance with both detectors, and the

Table 2: **ImageNet-1k validation accuracy.** The GFLOPs are measured with the specific resolution. Models with a comparable number of parameters are grouped together.

| Method | Type | #Params.(M) | Input | #FLOPs | Top-1 (%) | Method | Type | #Params.(M) | Input | #FLOPs | Top-1 (%) |
|---|---|---|---|---|---|---|---|---|---|---|---|
| ShuffleNet v2-50 | CNN | 2.3 | $224^2$ | 2.3G | 77.2 | VanillaNet-6 | CNN | 33.0 | $224^2$ | 6.0G | 82.9 |
| Mobile-Former | Trans | 4.6 | $224^2$ | 1.2G | 72.8 | XCiT-S12/16 | Trans | 26.0 | $224^2$ | 4.8G | 82.0 |
| MobileViT-S | Trans | 5.6 | $256^2$ | 1.8G | 78.4 | GroupMixFormer-S | Trans | 22.4 | $224^2$ | 5.2G | **83.4** |
| GroupMixFormer-M | Trans | 5.7 | $224^2$ | 1.4G | **79.6** | GroupMixFormer-S | Trans | 22.4 | $384^2$ | 15.2G | **85.0** |
| GroupMixFormer-M | Trans | 5.7 | $384^2$ | 4.0G | **81.5** | ResNet101 | CNN | 44.7 | $224^2$ | 7.9G | 77.4 |
| GroupMixFormer-M | Trans | 5.7 | $448^2$ | 5.4G | **81.8** | ResNeXt101-32x4d | CNN | 44.2 | $224^2$ | 8.0G | 78.8 |
| ResNet18 | CNN | 11.7 | $224^2$ | 1.8G | 69.8 | ConvNeXt-S | CNN | 50.0 | $224^2$ | 8.7G | 83.1 |
| PVT-Tiny | Trans | 13.2 | $224^2$ | 1.9G | 75.1 | ConvNeXt-B | CNN | 89.0 | $224^2$ | 15.4G | 83.8 |
| PVTv2-B1 | Trans | 13.1 | $224^2$ | 2.1G | 78.7 | ConvNeXt-L | CNN | 198.0 | $224^2$ | 34.4G | 84.3 |
| CoaT Mini | Trans | 10.0 | $224^2$ | 6.8G | 81.0 | PVT-Large | Trans | 61.4 | $224^2$ | 9.8G | 81.7 |
| EffNet-B4 | CNN | 19.0 | $224^2$ | 4.2G | 82.9 | PVTv2-B3 | Trans | 45.2 | $224^2$ | 6.9G | 83.2 |
| GroupMixFormer-T | Trans | 11.0 | $224^2$ | 3.7G | **82.6** | Swin-B | Trans | 88.0 | $224^2$ | 15.4G | 83.5 |
| GroupMixFormer-T | Trans | 11.0 | $384^2$ | 10.9G | **84.1** | Swin-B | Trans | 88.0 | $384^2$ | 47.0G | 84.5 |
| GroupMixFormer-T | Trans | 11.0 | $448^2$ | 14.9G | **84.3** | CSWin-B | Trans | 23.0 | $224^2$ | 4.3G | 82.7 |
| ResNet50 | CNN | 25.6 | $224^2$ | 4.1G | 76.5 | MViTv2-B | Trans | 78.0 | $224^2$ | 15.0G | 84.2 |
| ResNeXt50-32x4d | CNN | 25.0 | $224^2$ | 4.3G | 77.6 | DaViT-B | Trans | 87.9 | $224^2$ | 15.5G | 84.6 |
| ConvNeXt-T | CNN | 29.0 | $224^2$ | 4.5G | 82.1 | CoaTLite Medium | Trans | 45.0 | $384^2$ | 28.7G | 84.5 |
| PVT-Small | Trans | 24.5 | $224^2$ | 3.8G | 79.8 | Focal-Small | Trans | 51.1 | $224^2$ | 9.1G | 83.5 |
| PVTv2-B2 | Trans | 25.4 | $224^2$ | 4.0G | 82.0 | Focal-Base | Trans | 89.8 | $224^2$ | 16.0G | 83.8 |
| Swin-T | Trans | 29.0 | $224^2$ | 4.5G | 81.3 | XCiT-M24/8 | Trans | 84.0 | $224^2$ | 63.9G | 83.7 |
| CoaT Small | Trans | 22.0 | $224^2$ | 12.6G | 82.1 | GroupMixFormer-B | Trans | 45.8 | $224^2$ | 17.6G | **84.7** |
| Focal-Tiny | Trans | 29.1 | $224^2$ | 4.9G | 82.2 | GroupMixFormer-B | Trans | 45.8 | $384^2$ | 51.6G | **85.8** |
| CSWin-T | Trans | 23.0 | $224^2$ | 4.3G | 82.7 | GroupMixFormer-L | Trans | 70.3 | $224^2$ | 36.1G | **85.0** |
| MViTv2-T | Trans | 24.0 | $224^2$ | 4.7G | 82.3 | GroupMixFormer-L | Trans | 70.3 | $384^2$ | 106.2G | **86.2** |
| DaViT-T | Trans | 28.3 | $224^2$ | 4.5G | 82.8 | | | | | | |

Table 3: **Object detection and instance segmentation on COCO 2017 (Lin et al., 2014) with Mask R-CNN (He et al., 2017) and RetinaNet (Lin et al., 2017).** All the models are pre-trained on ImageNet-1K (Russakovsky et al., 2015). 'P' represents the number of parameters, and 'MS' denotes multi-scale training. The 3x schedule strictly follows (Chen et al., 2019).

| Backbone | #P (M) | Mask R-CNN 3× + MS | | | | | | RetinaNet 3× + MS | | | | | |
|---|---|---|---|---|---|---|---|---|---|---|---|---|---|
| | | $AP^b$ | $AP^b_{50}$ | $AP^b_{75}$ | $AP^m$ | $AP^m_{50}$ | $AP^m_{75}$ | $AP^b$ | $AP^b_{50}$ | $AP^b_{75}$ | $AP^b_S$ | $AP^b_M$ | $AP^b_L$ |
| ResNet18 (He et al., 2016) | 31.2/21.3 | 36.9 | 57.1 | 40.0 | 33.6 | 53.9 | 35.7 | 35.4 | 53.9 | 37.6 | 19.5 | 38.2 | 46.8 |
| PVT-Tiny (Wang et al., 2021) | 32.9/23.0 | 39.8 | 62.2 | 43.0 | 37.4 | 59.3 | 39.9 | 39.4 | 59.8 | 42.0 | 25.5 | 42.0 | 52.1 |
| CoaT Mini (Xu et al., 2021) | 30.0/– | 46.5 | 67.9 | 50.7 | 41.8 | 65.3 | 44.8 | – | – | – | – | – | – |
| CoaT-Lite Mini (Xu et al., 2021) | 31.0/– | 42.9 | 64.7 | 46.7 | 38.9 | 61.6 | 41.7 | – | – | – | – | – | – |
| GroupMixFormer-T | 30.8/20.2 | **47.5** | 68.9 | 52.2 | **42.4** | 66.1 | 45.9 | **46.3** | 67.6 | 49.4 | 32.0 | 50.3 | 59.9 |
| ResNet50 (He et al., 2016) | 44.2/37.7 | 41.0 | 61.7 | 44.9 | 37.1 | 58.4 | 40.1 | 39.0 | 58.4 | 41.8 | 22.4 | 42.8 | 51.6 |
| PVT-Small (Wang et al., 2021) | 44.1/34.2 | 43.0 | 65.3 | 46.9 | 39.9 | 62.5 | 42.8 | 42.2 | 62.7 | 45.0 | 26.2 | 45.2 | 57.2 |
| Swin-T (Liu et al., 2021b) | 48.0/39.0 | 46.0 | 68.1 | 50.3 | 41.6 | 65.1 | 44.9 | 45.0 | 65.9 | 48.4 | 29.7 | 48.9 | 58.1 |
| Focal-Tiny (Yang et al., 2021) | 48.8/39.4 | 47.2 | 69.4 | 51.9 | 42.7 | 66.5 | 45.9 | 45.5 | 66.3 | 48.8 | 31.2 | 49.2 | 58.7 |
| CoaT S (Xu et al., 2021) | 42.0/– | 49.0 | 70.2 | 53.8 | 43.7 | 67.5 | 47.1 | – | – | – | – | – | – |
| CoaT-Lite S (Xu et al., 2021) | 40.0/– | 45.7 | 67.1 | 49.8 | 41.1 | 64.1 | 44.0 | – | – | – | – | – | – |
| Uniformer-$S_{h14}$ (Li et al., 2022) | 41.0/– | 48.2 | 70.4 | 52.5 | 43.4 | 67.1 | 47.0 | – | – | – | – | – | – |
| GroupMixFormer-S | 42.2/31.9 | **49.1** | 70.2 | 53.7 | **43.5** | 67.4 | 47.3 | **47.6** | 68.5 | 51.3 | 33.1 | 51.2 | 61.3 |
| ResNet101 (He et al., 2016) | 63.2/56.7 | 42.8 | 63.2 | 47.1 | 38.5 | 60.1 | 41.3 | 40.9 | 60.1 | 44.0 | 23.7 | 45.0 | 53.8 |
| ResNeXt101-32x4d (Xie et al., 2017) | 62.8/56.4 | 44.0 | 64.4 | 48.0 | 39.2 | 61.4 | 41.9 | 41.4 | 61.0 | 44.3 | 23.9 | 45.5 | 53.7 |
| ResNeXt101-64x4d (Xie et al., 2017) | 101.9/95.5 | 44.4 | 64.9 | 48.8 | 39.7 | 61.9 | 42.6 | 41.8 | 61.5 | 44.4 | 25.2 | 45.4 | 54.6 |
| PVT-Medium (Wang et al., 2021) | 63.9/53.9 | 44.2 | 66.0 | 48.2 | 40.5 | 63.1 | 43.5 | 43.2 | 63.8 | 46.1 | 27.3 | 46.3 | 58.9 |
| PVT-Large (Wang et al., 2021) | 81.0/71.1 | 44.5 | 66.0 | 48.3 | 40.7 | 63.4 | 43.7 | 43.4 | 63.6 | 46.1 | 26.1 | 46.0 | 59.5 |
| Swin-S (Liu et al., 2021b) | 69.0/60.0 | 48.5 | 70.2 | 53.5 | 43.3 | 67.3 | 46.6 | 46.4 | 67.0 | 50.1 | 31.0 | 50.1 | 60.3 |
| Swin-B (Liu et al., 2021b) | 107.0/98.0 | 48.5 | 69.8 | 53.2 | 43.4 | 66.8 | 49.6 | 45.8 | 66.4 | 49.1 | 29.9 | 49.4 | 60.3 |
| Focal-Small (Yang et al., 2021) | 71.2/61.7 | 48.8 | 70.5 | 53.6 | 43.8 | 67.7 | 47.2 | 47.3 | 67.8 | 51.0 | 31.6 | 50.9 | 61.1 |
| Focal-Base (Yang et al., 2021) | 110.0/100.8 | 49.0 | 70.1 | 53.6 | 43.7 | 67.6 | 47.0 | 46.9 | 67.8 | 50.3 | 31.9 | 50.3 | 61.5 |
| Uniformer-$B_{h14}$ (Li et al., 2022) | 69.0/– | 50.3 | 72.7 | 55.3 | 44.8 | 69.0 | 48.3 | – | – | – | – | – | – |
| GroupMixFormer-B | 65.6/55.5 | **51.5** | 72.7 | 56.8 | **45.9** | 70.0 | 50.0 | **50.2** | 71.7 | 55.3 | 36.4 | 52.1 | 62.3 |

consistent and significant improvements demonstrate the effectiveness of the Group-Mix mechanism, which is found to be able to capture the fine-grained features to facilitate dense predictions.

**Semantic Segmentation.** Table 3 also shows the semantic segmentation results on COCO with Mask-RCNN. Our GroupMixFormer-T achieves an $AP^m$ of 42.4%, 0.6% higher than Coat Mini and 1.7% higher than PVT-Large, which is impressive. Our GroupMixFormer-B performs 1.1% better than Uniformer-B (i.e., 45.9% v.s. 44.8%). On ADE20K, we use UperNet and Semantic FPN and report the results in Table 4. Similarly, we observe that GroupMixFormers consistently achieves favorable performance compared to the existing backbones. For example, GroupMixFormer-T, though much smaller, performs 2.0% better than XCiT-S12/8 (i.e., 46.2% v.s. 44.2%, 14.1 M v.s. 30.4 M) with Semantic FPN. Notably, GroupMixFormer-T outperforms XCiT-M24/16 by 0.3%, though the latter is 6.4× as big as GroupMixFormer-T (i.e., 46.2% v.s. 45.9%, 14.1 M v.s. 90.8 M). Similarly, with UperNet, GroupMixFormers perform much better than the other bigger models, showing a clearly better trade-off between performance and efficiency. Such significant improvements suggest that the Group-Mix mechanism is able to produce high-quality features for pixel-level predictions.

Table 4: **Semantic segmentation on ADE20k (Zhou et al., 2019) with UperNet (Xiao et al., 2018) and Semantic FPN (Kirillov et al., 2019).** All the models are pre-trained on ImageNet-1K (Russakovsky et al., 2015) and finetuned with task-specific heads. We follow the standard training and evaluation processes in (Contributors, 2020) for fair comparisons.

| Backbone | Semantic FPN | | UperNet | | Backbone | Semantic FPN | | UperNet | |
|---|---|---|---|---|---|---|---|---|---|
| | #Param(M) | mIoU(%) | #Param(M) | mIoU(%) | | #Param(M) | mIoU(%) | #Param(M) | mIoU(%) |
| ResNet18 | 15.5 | 32.9 | – | – | ResNet101 | 47.5 | 38.8 | 85.5 | 43.8 |
| PVT-Tiny | 17.0 | 35.7 | – | – | ResNeXt101-32x4d | 47.1 | 41.6 | – | – |
| XCiT-T12/16 | 8.4 | 38.1 | 33.7 | 41.5 | ResNeXt101-64x4d | 65.1 | 44.8 | – | – |
| XCiT-T12/8 | 8.4 | 39.9 | 33.7 | 43.5 | PVT-Large | 65.1 | 42.1 | – | – |
| GroupMixFormer-T | 14.1 | 46.2 | 39.1 | 47.4 | Swin-S | – | – | 81.0 | 47.6 |
| ResNet50 | 28.5 | 36.7 | 66.5 | 42.0 | Swin-B | – | – | 121.0 | 48.1 |
| PVT-Small | 28.2 | 39.8 | – | – | Focal-S | – | – | 85.0 | 50.0 |
| Swin-T | – | – | 59.9 | 44.5 | Focal-B | – | – | 126.0 | 50.5 |
| Focal-T | – | – | 62 | 45.8 | XCiT-M24/16 | 90.8 | 45.9 | 109.0 | 47.6 |
| XCiT-S12/16 | 30.4 | 43.9 | 52.4 | 45.9 | XCiT-M24/8 | 90.8 | 46.9 | 109.0 | 48.4 |
| XCiT-S12/8 | 30.4 | 44.2 | 52.4 | 46.6 | GroupMixFormer-B | 49.7 | 50.0 | 75.5 | 51.2 |
| GroupMixFormer-S | 26.3 | 47.8 | 51.1 | 49.6 | | | | | |

Table 5: **Ablation studies on the group aggregators in GMA Block.** We use $Agg^1$, $Agg^2$, and $Agg^3$ to denote aggregators (in the pre-attention branch) with kernel sizes of 3, 5, and 7, respectively, and $Agg^0$ to denote the aggregator (in the non-attention branch) as shown in Figure 3. We report the Top-1 accuracy on ImageNet-1k together with $AP^m$ and $AP^b$ on COCO.

| Method | $Agg^0$ | $Agg^1(3\times3)$ | $Agg^2(5\times5)$ | $Agg^3(7\times7)$ | #Params.(M) | #GFLOPs | Top-1 (%) | $AP^b$ | $AP^m$ |
|---|---|---|---|---|---|---|---|---|---|
| GroupMixFormer-T | ✗ | ✗ | ✗ | ✗ | 10.5 | 3.4 | 80.9 | 43.9 | 39.6 |
| GroupMixFormer-T | ✗ | ✓ | ✓ | ✓ | 10.8 | 3.5 | 81.9 (+1.0) | 44.6 | 40.4 |
| GroupMixFormer-T | ✓ | ✗ | ✗ | ✗ | 10.7 | 3.5 | 81.3 (+0.4) | 44.1 | 40.3 |
| GroupMixFormer-T | ✓ | ✓ | ✗ | ✗ | 10.8 | 3.6 | 82.2 (+1.3) | 44.6 | 40.3 |
| GroupMixFormer-T | ✓ | ✗ | ✓ | ✗ | 10.8 | 3.6 | 82.3 (+1.4) | 44.6 | 40.4 |
| GroupMixFormer-T | ✓ | ✗ | ✗ | ✓ | 10.9 | 3.6 | 82.3 (+1.4) | 44.8 | 40.3 |
| GroupMixFormer-T | ✓ | ✓ | ✓ | ✓ | 11.0 | 3.7 | 82.6 (+1.7) | 45.4 | 40.6 |

## 4.3 ABLATION STUDIES

We conduct ablation studies to analyze the key designs of GroupMixFormer. (1) We first analyze the necessity of the aggregators by changing the structural designs of GMA. (2) We experiment with various implementations of aggregators to see if other sliding-window-based operations, except for convolution, also work. (3) We conduct experiments to verify that GMA is not merely a trivial combination of convolution and self-attention. (4) We plug GMA Blocks into the other popular ViT architectures to verify if the superior performance of GroupMixFormer is merely due to the architectural designs (e.g., overlapping embedding layers and numbers of blocks within each stage). For image classification, we train GroupMixFormer-T for 300 epochs on ImageNet-1k ($224^2$) and test with the validation set. For object detection and semantic segmentation, we train Mask R-CNN with the $1\times$ schedule (Chen et al., 2019) on COCO.

**Group aggregators are necessary.** Table 5 shows the results of ablating the aggregators. We first construct a GroupMixFormer-T baseline by replacing all of the five branches in GMA Blocks with identity mappings, so that the block degrades into a regular self-attention module. In the first group of experiments, we restore the aggregators in the non-attention branch ($Agg^0$) or the three pre-attention branches ($Agg^1$, $Agg^2$ and $Agg^3$). Every model is trained from scratch with the same configurations as described in Section 4.1. It could be observed that the aggregators are all critical, as they improve the top-1 accuracy by 0.4% and 1.0%, respectively.

Moreover, the second group of experiments in Table 5 shows that using aggregators in all of the three pre-attention branches yields better performance than using any single one. Similar experimental results are observed in object detection and semantic segmentation as well. Using all the aggregators improves the baseline performance by a certain margin (e.g., +0.7% $AP^b$ and +0.5% $AP^m$). These results indicate that modeling correlations in a more comprehensive manner is able to provide fine-grained visual representations to benefit dense prediction scenarios.

Table 6: ImageNet-1k classification, COCO det and instance seg (1x with Mask RCNN) performances on various aggregators.

| Method | Implementation | Top-1 (%) | $AP^b$ | $AP^m$ |
|---|---|---|---|---|
| GroupMixFormer-T | MinPool | 82.3 | 42.4 | 39.7 |
| GroupMixFormer-T | MaxPool | 82.2 | 42.3 | 39.7 |
| GroupMixFormer-T | AvgPool | 82.2 | 42.3 | 39.6 |
| GroupMixFormer-T | DWConv | 82.6 | 42.5 | 39.8 |

Table 7: ImageNet-1k validation of replacing GMA with other attention modules on GroupMixFormer-T.

| Attention Type | #Params.(M) | #GFLOPs | Top-1 (%) |
|---|---|---|---|
| GMA | 10.5 | 3.4 | 82.6 |
| Swin-attention | 10.8 | 3.5 | 79.9 (-2.7) |
| PVT-attention | 16.3 | 3.3 | 79.1 (-3.5) |

We then analyze the impact of various kernel sizes of pre-attention aggregators on performance. Without altering the non-attention branch, we replace all of the pre-attention aggregators with either $Agg^1$ (3×3 convolution), $Agg^2$ (5×5) or $Agg^3$ (7×7). The second set of results in Table 5 indicates that the utilization of any group aggregators enhances classification

Table 8: ImageNet-1k validation accuracy of incorporating aggregators to other ViT architectures.

| Structures | Aggregators | #Params.(M) | #GFLOPs | Top-1 (%) |
|---|---|---|---|---|
| Swin-T | ✗ | 28.3 | 4.5 | 80.8 |
| Swin-T | ✓ | 28.8 | 4.8 | 81.5 (+0.7) |
| PVT-Small | ✗ | 24.5 | 3.8 | 79.8 |
| PVT-Small | ✓ | 25.2 | 4.0 | 80.6 (+0.8) |

and dense prediction performance, with a diverse combination of 3×3, 5×5, and 7×7 yielding the most optimal results. Specifically, GroupMixFormer-T equipped with diverse aggregators outperforms the baseline by +1.7% classification accuracy, +1.5% $AP^b$ in object detection, and +1.0% $AP^m$ in semantic segmentation, which suggests that modeling the correlations among groups of *diverse* sizes is the key to performance boost.

**Depthwise Convolutions are effective aggregators.** Note that the implementations of aggregators Agg(·) could be various. Table 6 shows our results regarding the effects of different aggregator implementations (e.g., depthwise convolution (Chollet, 2017), max-pooling, or average-pooling). It's empirically observed that the aggregators implemented by depthwise convolution achieve the best performance (82.6% Top-1 accuracy on classification, 42.5% $AP^b$ for detection, and 39.7% $AP^m$ for instance segmentation with Mask R-CNN ). Compared with the max-pooling and min-pooling operations, convolutional aggregators take advantage of involving more learnable parameters for computing correlations, thus achieving better performances.

**Performance gains are not derived from macro-structures.** Compared with the representative works (Liu et al., 2021b; Wang et al., 2021; Touvron et al., 2021), our GroupMixFormer is deeper and has different implementations of patch embedding. In order to justify that the performance gains are not simply due to a better combination of architectural hyper-parameters (including the dimensions of tokens, expansion ratios, and layer depths as introduced in Table 1), we replace the GMA Blocks in GroupMixFormer-T with the Swin-attention or PVT-attention. The results in Table 7 show that simply replacing the GMA causes a significant performance drop, which justifies that the performance gain is due to the advanced attention mechanism instead of the architecture.

**GMA is not merely a trivial combination of convolution and self-attention.** We conduct further experiments to validate that our proposed GroupMixFormer is essentially different from a simple combination of convolution and self-attention. Specifically, we remove all the group aggregators from GroupMixFormer-T and insert a group of convolutional layers organized in the same manner (i.e., a combination of parallel identity mapping, 3×3, 5×5 and 7×7 layers) before the whole self-attention module. The accuracy drops by 1.1% in the Top-1 accuracy (81.5% v.s. 82.6%).

**Aggregator is an advanced universal building block that could be applied to the other ViTs.** We may also incorporate aggregators into representative ViTs (e.g., Swin (Liu et al., 2021b) and PVT (Wang et al., 2021)) by simply inserting the them into their original attention modules to process their Query, Key, and Value. The results in Table 8 show that such the strategy generally boosts ViTs by a clear margin. For example, Swin-T with aggregators achieves 81.5% Top-1 accuracy, which is 0.7% higher than its original result. It indicates that the proposed aggregators advances ViTs by modeling the group correlations and thus leading to a comprehensive understanding of the tokens.

## 5 CONCLUSION

In this paper, we proposed an advanced attention mechanism, named Group-Mix Attention (GMA). In contrast to the popular multi-head self-attention (MHSA) that only models the correlations among individual tokens, the proposed GMA utilizes the group aggregators to simultaneously capture the token-to-token, token-to-group, and group-to-group correlations. We proposed GroupMixFormer based on GMA and instantiated a series of practical visual backbones with different sizes. Extensive experiments on the standard visual recognition benchmarks (including image classification, object detection, and semantic segmentation) have validated the effectiveness of the proposed GMA and GroupMixFormer.

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

# Appendix

## CONTENTS

# Appendix

This appendix includes detailed illustrations of the algorithm, training configurations and additional experiments. In Algorithm 1, we present the PyTorch-style pseudocode of GMA Block for easy implementation. In Appendix A, we detail the attention computation and elaborate on the training configurations for image classification, object detection, and instance/semantic segmentation. Besides, in Appendix D and Appendix E, we present additional experiments and visualizations to further validate GroupMixFormer's effectiveness, respectively. The core cored are provided in the supplementary material. We will make the code and trained models publicly available.

---

**Algorithm 1** PyTorch-style Pseudocode of GMA Block.

```
# x: the input token with shape of (B, N, D), B is batch size, N=H*W, D is dimension
# qkv_mapping(): linear mapping (in=D, out=D*3) to generate Q, K, V
# att(): efficient multi-head Q-K-V computation
# token_ensemble(): linear mapping (in=out=D) to combine the outputs from the attention
    and non-attention branches
# act: activation function, implemented by HardSwish
# norm: normalization function, implemented by LayerNorm
# The aggregator is implemented by a depth-wise convolution (channels=groups=D//5)
    following a linear mapping
def GMA(x):
    B,N,D=x.shape
    split_dim = D//5

    # Generate Q/K/V
    qkv = qkv_mapping(x).reshape(B, N, 3, D).permute(2, 0, 1, 3).reshape(3*B, N, D)
    qkv = qkv.transpose(1, 2).view(3*B, D, H, W)
    qkv = qkv.split([split_dim]*5, dim=1)
    # Now qkv[i] is the i-th branch with shape of (3*B, split_dim, H, W)

    qkv_pre_att_0 = act(norm(qkv[0]))
    # Generate group proxies via different aggregators
    qkv_pre_att_1 = act(norm(aggregator_pre_att_3x3(qkv[1])))
    qkv_pre_att_2 = act(norm(aggregator_pre_att_5x5(qkv[2])))
    qkv_pre_att_3 = act(norm(aggregator_pre_att_7x7(qkv[3])))

    # Non-attention branch
    qkv_non_att = qkv[4].reshape(3, B, split_dim, H, W).permute(1, 0, 2, 3, 4).reshape(B,
        3*split_dim, H, W)
    x_non_att = act(norm(aggregator_non_att_3x3(qkv_non_att)).reshape(B, split_dim, H, W))

    # Efficient multi-head Q-K-V self-Attention. We ignore the number of heads for brevity
    # Its input is (3*B, D*4/5, H, W), output is (B, D*4/5, H, W)
    qkv_input = torch.cat([qkv_pre_att_0, qkv_pre_att_1, qkv_pre_att_2, qkv_pre_att_3],
        dim=1)
    x_att = att(qkv_input)

    # combine the outputs from attention and the non-attention branch
    x = torch.cat([x_att, x_non_att], dim=1) # the shape becomes (B, D, H, W)
    x = x.reshape(B, D, N).permute(0, 2, 1) # the shape becomes (B, N, D)
    x = token_ensemble(x)
    return x
```

---

## A  IMPLEMENTATION DETAILS

### A.1  ATTENTION COMPUTATION

We detail the attention computation adopted by GroupMixFormer in this appendix. The Q/K/V entries are first uniformly divided into five segments where we perform group aggregation on four segments. We use $X_i^q, X_i^k, X_i^v$ ($i \in [1, 2, 3, 4]$) to denote the segments divided from Q/K/V entries, respectively. To produce the group proxies $Q', K'$, and $V'$, we first employ the aggregation operation on the segments as $\text{Agg}^i(X_i^q)$, $\text{Agg}^i(X_i^k)$ and $\text{Agg}^i(X_i^v)$. Then we concatenate all the four ($i \in [1, 2, 3, 4]$) aggregated features to output group proxies $Q', K'$, and $V'$. Afterward, we perform attention computation as introduced in (Xu et al., 2021; Ali et al., 2021; Shen et al., 2021) on the group proxies to generate

the final output Att.

$$\text{Att} = \frac{Q'}{\sqrt{d}} \text{Softmax}(K'^T V').$$

## A.2 IMAGE CLASSIFICATION

The standard ImageNet-1K dataset (Russakovsky et al., 2015) contains about 1.3 million training samples and 50K validation samples from 1000 categories. We experiment with input resolutions of $224 \times 224$, $384 \times 384$, or $448 \times 448$. We follow (Touvron et al., 2021) for data augmentation, including Mixup (Zhang et al., 2017), CutMix (Yun et al., 2019), random erasing (Zhong et al., 2020), etc. We use the same training recipes as (Liu et al., 2021b). For training with $224 \times 224$, all GroupMixFormer instances are trained for 300 epochs with a batch size of 1024. The initial learning rate is set to $10^{-3}$ with a linear warm-up for 20 epochs and then cosine annealing towards zero. We adopt the AdamW optimizer (Loshchilov & Hutter, 2017) with a weight decay coefficient of 0.05. The drop-path rates (Huang et al., 2016) are set to 0.0, 0.1, 0.2, 0.4 and 0.5 for GroupMixFormer-M/T/S/B/L, respectively. Besides, for higher resolutions (i.e., $384 \times 384$ and $448 \times 448$), we finetune the $224 \times 224$-pretrained models for another 30 epochs with an initial learning rate of $2 \times 10^{-6}$ and a linear warm-up for 5 epochs and then cosine annealing. For finetuning, we use AdamW optimizer (Loshchilov & Hutter, 2017) with a weight decay coefficient of $1.0 \times 10^{-8}$.

## A.3 OBJECT DETECTION AND INSTANCE SEGMENTATION

For object detection, we experiment on COCO 2017 (Lin et al., 2014) with Mask R-CNN (He et al., 2017) and RetinaNet (Lin et al., 2017). All models are initialized with the weights pretrained on ImageNet-1K (Russakovsky et al., 2015). The detectors are finetuned on COCO train2017 (118k images) and evaluated on COCO val2017 (5k images). For data augmentation, we adopt multi-scale training as a common practice (Liu et al., 2021b). We also follow the standard $3 \times$ (36-epoch) training schedules provided in (Chen et al., 2019). We use AdamW (Loshchilov & Hutter, 2017) with a weight decay coefficient of 0.05 for Mask R-CNN and $10^{-4}$ for RetinaNet.

For instance segmentation, we benchmark GroupMixFormer models on COCO 2017 (Lin et al., 2014) with Mask R-CNN (He et al., 2017) with the same configurations as described above.

Moreover, we present additional results with Cascade Mask R-CNN (Cai & Vasconcelos, 2019) in this supplementary material. We use the same training configurations as Mask R-CNN.

## A.4 SEMANTIC SEGMENTATION

For semantic segmentation, we experiment on ADE20k (Zhou et al., 2019) with UperNet (Xiao et al., 2018) and Semantic FPN (Kirillov et al., 2019)). ADE20K contains $\sim$20k, $\sim$2k, and $\sim$3k images for training, validation, and testing, respectively, from 150 categories. Following common practices (Wang et al., 2021; Liu et al., 2021b), we randomly resize and crop the image to $512 \times 512$ for training, and rescale the shorter side to 512 pixels for testing. We use AdamW with a weight decay coefficient of $10^{-4}$ for Semantic FPN and 0.01 for UperNet. The Semantic FPN is trained for 80k iterations while the UperNet is trained for 160k iterations. The learning rate is initialized as $6 \times 10^{-5}$, warmed up linearly in 1500 iterations, and then decayed following the polynomial decay schedule with a power of 0.9.

# B SPEED ANALYSIS

We empirically found that implementing the aggregators in GMA with DW-Conv indeed slow-down the inference speed. For instance, as shown in Table 9, when tested on the single V100 GPU, our throughput (596 images/s) is smaller than the prevalent backbones (e.g., Swin-T with 755 images/s, CSWin-T with 701 images/s). However, our model outperforms others by large margins in recognition performance. Besides, it's noteworthy that with accuracy maintained, the speed of GroupMixFormercould be further improved by implementing with more efficient aggregators (e.g., +15 image/s by AvgPool as shown in Table 9).

Table 9: Comparisons on inference speed with different models.

| Method | Swin-T | PVT-S | CSWin-T | GroupMixFormer-S | GroupMixFormer-S(AvgPool) |
|---|---|---|---|---|---|
| Throughput (images/s) | 755 | 820 | 701 | 596 | 611 |
| #Param.(M) | 29.0 | 24.5 | 23.0 | 22.4 | 22.1 |
| #FLOPs.(G) | 4.5 | 3.8 | 4.3 | 5.2 | 5.0 |
| Performance (%) | 81.3 | 79.8 | 82.7 | 83.4 | 83.0 |

## C  OPTIMAL CONFIGURATIONS ON KERNEL SIZES OF AGGREGATORS

To find the optimal configuration, we undertake two approaches: (1) enlarging the kernel size, and (2) altering the kernel configurations in varying orders. The first approach entails increasing the kernel sizes from (3,5,7) to (5,7,9). For the second approach, we deploy aggregators with larger kernels in the shallow layers and smaller kernels in the deeper layers, as well as in a reversed configuration. However, as demonstrated in  Table 10, neither of these modifications proved as effective as the configuration we ultimately adopted.

Table 10: Explorations on optimal kernel configurations with GroupMixFormer-T

| Strategy | #Params.(M) | #GFLOPS | Top-1 Acc (%) |
|---|---|---|---|
| kernel sizes = [5,7,9] | 11.2 | 3.9 | 82.0 |
| large kernel to small kernel | 10.8 | 3.7 | 82.2 |
| small kernel to large kernel | 11.0 | 3.7 | 82.0 |
| Ours | 11.0 | 3.7 | **82.6** |

## D  ADDITIONAL RESULTS WITH CASCADE MASK R-CNN

To further verify the effectiveness of our proposed model, we equip GroupMixFormer with a more powerful object detector, i.e., Cascaded Mask R-CNN (Cai & Vasconcelos, 2019). Detailed implementations are presented in Appendix A.3. Results in Table 11 show GroupMixFormer consistently outperforms the prevalent Transformer-based backbones (e.g., PVT-V2 (Wang et al., 2022) and Swin (Liu et al., 2021b)). Besides, with fewer parameters (68.6M *v.s.* 86.7M), our GroupMixFormer-T obtains a comparable performance with Focal-T (around 51.5% $AP^b$). Our GroupMixFormer-S achieves new state-of-the-art performance with an $AP^b$ of 51.9%.

Table 11: **Object detection and instance segmentation performance** on COCO 2017 (Lin et al., 2014) with Cascade Mask R-CNN (Cai & Vasconcelos, 2019).

| Backbone | Cascade Mask R-CNN 3× + MS | | | | | | |
|---|---|---|---|---|---|---|---|
| | #P (M) | $AP^b$ | $AP^b_{50}$ | $AP^b_{75}$ | $AP^m$ | $AP^m_{50}$ | $AP^m_{75}$ |
| ResNet50 (He et al., 2016) | 82.0 | 46.3 | 64.3 | 50.5 | – | – | – |
| PVTv2-b2-Linear (Wang et al., 2022) | 80.1 | 50.9 | 69.5 | 55.2 | 44.0 | 66.8 | 47.7 |
| PVTv2-b2 (Wang et al., 2022) | 82.9 | 51.1 | 69.8 | 55.3 | 44.4 | 67.2 | 48.1 |
| Swin-T (Liu et al., 2021b) | 85.6 | 50.2 | 68.8 | 54.7 | 43.5 | 66.1 | 46.9 |
| Focal-T (Xu et al., 2021) | 86.7 | 51.5 | 70.6 | 55.9 | – | – | – |
| GroupMixFormer-T (ours) | 68.6 | **51.5** | 70.2 | 55.7 | **44.4** | 67.5 | 48.2 |
| GroupMixFormer-S (ours) | 80.0 | **51.9** | 70.7 | 56.1 | **45.1** | 68.3 | 48.4 |

## E  ATTENTION VISUALIZATION

We present attention response maps in Figure 4. We show input images in (a), and the attention response maps from the ensemble layer in (b). Besides, the response maps of the outputs from the pre-attention branches and non-attention branch are shown in (c) to (g), respectively. We observe that applying self-attention on individual tokens sometimes fails to attend to the object, as shown in (c). In such a case, calculating the correlations among the group proxies, which are generated by the aggregators, may help. For example, as shown in the third row, calculating correlations among

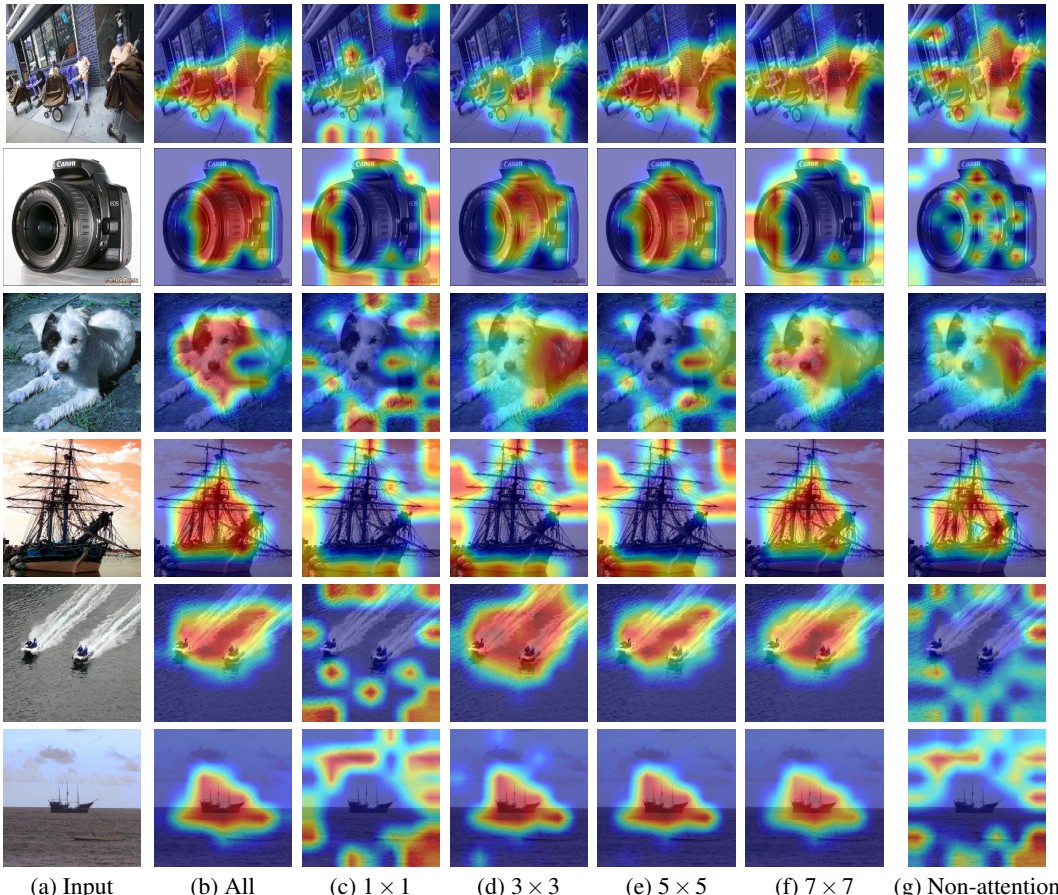

|  (a) Input | (b) All | (c) $1 \times 1$ | (d) $3 \times 3$ | (e) $5 \times 5$ | (f) $7 \times 7$ | (g) Non-attention |

Figure 4: **Attention visualizations on GroupMixFormer-S.** The model attends to the pixels marked as red more than the others. Input images are shown in (a). In (c) to (f), we show the attention response maps from different aggregators in the pre-attention branches. In (g), we show the response maps from the aggregators in the non-attention branch. The combined response maps (outputs from the token ensemble layer) are shown in (b).

the groups, which are processed by aggregators with kernel sizes of 3 and 7, succeed in focusing on the dog, while modeling the token-to-token correlations in (c) focuses more on the background. These results indicate that there exist some patterns so that some tokens should be handled as a whole to capture the object features. In GMA, the representations captured by different aggregators are combined. It validates that comprehensively modeling the token-to-token, token-to-group, and group-to-group correlations leads to better vision recognition.

