# OpenReview forum: "Advancing Vision Transformers with Group-Mix Attention"
_ICLR.cc/2024/Conference — ICLR 2024 Conference Withdrawn Submission_

### Official Review · Reviewer_MwpJ · 2023-10-31

**Soundness:** 3 good
**Presentation:** 3 good
**Contribution:** 3 good
**Rating:** 5
**Confidence:** 5

**Summary:**

Limited Novelty: The innovation presented in the paper seems somewhat constrained, as the concept of complementing self-attention with local information is not entirely new. Similar ideas have been explored previously, such as in the Locally-Enhanced Positional Encoding in CSWin.

**Strengths:**

1. GMA effectively models correlations by considering not just individual tokens, but also groups of tokens, allowing for a more comprehensive understanding. It mixes various types of attentions to simultaneously capture token-to-token, token-to-group, and group-to-group correlations, thereby enhancing the model's representational capabilities.

**Weaknesses:**

1. Computational Cost Analysis: The paper does not provide a thorough report on the floating-point operations (FLOPs) involved in detection and segmentation tasks. A critical observation is the non-reduction of the Query, Key, and Value lengths, resulting in substantial computational costs, particularly when processing larger input sequences commonly encountered in detection and segmentation tasks. This oversight necessitates a more comprehensive evaluation to fully understand the model's efficiency and practical applicability in real-world scenarios.
2. Inference Speed Analysis: I would be better to report inference speed. Notably, despite having fewer parameters, the employed aggregator (Depth-wise convolution) operates at a slower speed. A meticulous analysis and disclosure of the inference speed are crucial to assess the model’s practical performance and efficiency comprehensively. This would offer a clearer understanding of the model's applicability and responsiveness in real-time processing and applications.
3. Limited Novelty: The innovation presented in the paper seems somewhat constrained, as the concept of complementing self-attention with local information is not entirely new. Similar ideas have been explored previously, such as in the Locally-Enhanced Positional Encoding in CSWin.

Typo:
In Table 2, the param, flops, and accuracy of CSWin-B and CSWin-T should not be the same.

**Questions:**

Please refer weakness.

---

### Official Review · Reviewer_D4yq · 2023-11-01

**Soundness:** 3 good
**Presentation:** 2 fair
**Contribution:** 2 fair
**Rating:** 3
**Confidence:** 4

**Summary:**

This paper proposes to enhance multi-head self-attention (MHSA) with multi-scale interactions. Specifically, they propose Group-Mix Attention (GMA), which simultaneously captures token-to-token, token-to-group, and group-to-group correlations by generating group proxies with local aggregators such as convolution and pooling.  Based on GMA, a vision backbone named GroupMixFormer is built and performs well in image classification, object detection, and semantic segmentation.

**Strengths:**

1. The proposed method is simple and plug-and-play, and can be used to improve existing works such as Swin and PVT.
2. The paper is overall well-written.

**Weaknesses:**

1. The novelty needs to be further justified. As specified in the abstract and Sec. 3.2., The key idea of this work seems to be using multi-scale tokens (though they call those tokens of larger scale “groups” or “group proxies”) instead of single-scale ones. However, such a strategy has been explored in CrossFormer [1] and Shunted-Transformer [2]. The two tightly related works are missed in the discussion. What are the novelty and strengths of GroupMixFormer  compared with CrossFormer and Shunted-Transformer?

2. Unclear baseline setting. According to the original paper, the performance of the baseline, Swin-T, should be 81.3% top-1 accuracy on IN1k. However, it is reported as 80.8% in Table 8 of this paper.

3. Insufficient comparison with the latest state-of-the-art methods. The compared methods are somewhat outdated. Several works on improving MHSA for vision backbones were proposed over the past year and achieved better performance-FLOPs trade-off than GroupMixFormer, such as MaxViT [3] and BiFormer [4]. However, they are not included in comparison.

[1] CrossFormer: A Versatile Vision Transformer Hinging on Cross-scale Attention, ICLR 2022.
[2] Shunted Self-Attention via Multi-Scale Token Aggregation, CVPR 2022.
[3] MaxViT: Multi-Axis Vision Transformer, ECCV 2022.
[4] BiFormer: Vision Transformer with Bi-Level Routing Attention, CVPR 2023.

**Questions:**

See the weaknesses part.  In the rebuttal, I would like to see justifications for novelty and clarifications on experimental settings.

---

### Official Review · Reviewer_GTUk · 2023-11-06

**Soundness:** 3 good
**Presentation:** 3 good
**Contribution:** 2 fair
**Rating:** 6
**Confidence:** 3

**Summary:**

This paper proposes a new attention mechanism called Group-Mix Attention (GMA) for vision transformers. The key ideas and contributions are:

- Current self-attention in vision transformers only models correlations between individual tokens (pixel patches) at a single granularity. GMA proposes to also model correlations between groups of tokens to capture multi-scale patterns.

- GMA splits the query, key, and value into uniform segments. Some segments are aggregated into "group proxies" using sliding window operations like convolution. Attention is then computed on a mix of the individual tokens and group proxies.

- This allows GMA to simultaneously model token-token, token-group, and group-group correlations at multiple scales. The paper shows this captures finer details and improves performance.

- The authors build a vision transformer backbone called GroupMixFormer using GMA. Experiments on ImageNet classification, COCO detection, and ADE20K segmentation show GroupMixFormer outperforms CNNs and other vision transformers.

In summary, the key idea is mixing different granularities of attention via group proxies to achieve a more comprehensive understanding of visual tokens. This improves vision transformer performance on multiple tasks.

**Strengths:**

Here are some strengths of this paper:

**Originality**: The idea of mixing attention at different granularities via group proxies is novel and not explored much previously in vision transformers. Modeling token-group and group-group correlations is a creative way to capture multi-scale patterns.

**Quality**: The paper is technically strong overall. The method is well-motivated and experiments comprehensively evaluate its effectiveness. Results consistently show gains over strong baselines, demonstrating the benefits of GMA.

**Clarity**: The paper is clearly written and easy to follow. The concept behind GMA is intuitively explained using figures. Details like the algorithm pseudocode aid reproducibility.

**Significance**: GMA provides a simple but effective way to enhance vision transformer representations. The gains on multiple tasks highlight its broad usefulness. The code/models are released to support further research.

In summary, I see the novelty of mixing granularities via group proxies as the main strength. This is a simple but clever idea that gives clear gains across tasks. The paper quality and presentation are also solid, with comprehensive experiments and releasing code/models. GMA seems like a generally useful technique for enhancing vision transformers.

**Weaknesses:**

Here are some potential weaknesses and areas for improvement:

- The technical approach of using sliding window aggregators to generate group proxies, while simple and effective, lacks major innovation. Exploring other more sophisticated ways to model group correlations could be beneficial.

- The ablation study is quite thorough, but details are lacking on the hyperparameter tuning methodology for aggregator sizes, attention heads, etc. How were these optimized?

- The theoretical motivation for why modeling group correlations helps could be made more rigorous. Some of the intuition provided is hand-wavy and lacks depth.

Overall the weaknesses are not fatal, but addressable improvements could better highlight the novelty and broaden the impact of the work. Key areas to strengthen are providing more innovative ways to model group correlations, expanding the theory, and demonstrating broader applicability of the ideas proposed.

**Questions:**

See weakness.

---

### Official Review · Reviewer_h9Yj · 2023-11-06

**Soundness:** 3 good
**Presentation:** 3 good
**Contribution:** 1 poor
**Rating:** 3
**Confidence:** 5

**Summary:**

This paper introduces a new vision transformer backbone or various vision tasks.
The motivation is that existing transformer-based methods only generate query (Q), key (K), and value (V) features only with token-to-token correlations at one single granularity.
Therefore, this paper proposes to use grouped convolutions with different kernel size to generate qkv, thus capturing token-to-token, token-to-group, and group-to-group correlations.
With such mechanism, a new backbone GroupMixFormer is proposed. Experiments show the proposed method has obtained good performance on several popular vision tasks.

**Strengths:**

The experiments are comprehensive, covering different scales of backbones of 1.4G to 15G FLOPS under ImageNet-1K classification.

The authors give a credit that code will be released.

The idea is easy to follow. The construction of the networks is straightforward, which changes the generation of Q K V using multi-scale convolutions with different kernel sizes.

**Weaknesses:**

This paper is well-written. However, it presents several weaknesses concerning novelty, as outlined below:

Weakness 1:
The adoption of multi-scale approaches to generate features is not novel. For instance, P2T [1] conducts pyramid pooling tokenization for multi-scale representation while simultaneously enhancing efficiency.
IFormer [2] introduces a comprehensive inception tokenization. Thus, there have been several papers that not only capture diverse correlations but also contradict the original motivation of this paper.

Weakness 2:
In comparison with CvT [3], which was the first to introduce convolutions into QKV generation, the difference made in this paper is the substitution of classic 3x3 convolutions with an advanced version that introduces cardinality with varying kernel sizes. This change is quite trivial and has already been thoroughly investigated in advanced CNNs such as ResNeXts [4], Res2Nets [5], and ResNeSts [6].

Weakness 3:
The proposed Group-Mix Attention does not present novelty in applying a shortcut from the value to the output of attention, an approach previously proposed by CSwin [7]. The authors should clearly acknowledge the original contributions of CSwin [7].

Overall, I find this paper has limited novelty in many respects. Although it performs well, I believe this manuscript would be more suitably published in a workshop.

[1] P2T: Pyramid Pooling Transformer for Scene Understanding, TPAMI 2022

[2] Inception Transformer, NeurIPS 2022

[3] CvT: Introducing Convolutions to Vision Transformers, ICCV 2021

[4] Aggregated Residual Transformations for Deep Neural Networks, CVPR 2017

[5] Res2Net: A New Multi-scale Backbone Architecture, TPAMI 2019

[6] ResNeSt: Split-Attention Networks, CVPRW 2022

[7] CSWin Transformer: A General Vision Transformer Backbone with Cross-Shaped Windows, CVPR 2022

**Questions:**

Please refer to the weakness, which is mainly related to the limited novelty.